# PER1 Oscillation in Rat Parathyroid Hormone and Calcitonin Producing Cells

**DOI:** 10.3390/ijms252313006

**Published:** 2024-12-03

**Authors:** Birgitte Georg, Henrik L. Jørgensen, Jens Hannibal

**Affiliations:** 1Department of Clinical Biochemistry, Bispebjerg University Hospital, 2400 Copenhagen, Denmark; jens.hannibal@regionh.dk; 2Department of Clinical Biochemistry, Amager and Hvidovre Hospital, 2650 Copenhagen, Denmark; hlj@dadlnet.dk; 3Department of Clinical Medicine, Faculty of Health and Medical Sciences, University of Copenhagen, 2200 Copenhagen, Denmark

**Keywords:** thyroid gland, parathyroid hormone, calcitonin, PER1, rat

## Abstract

Many endocrine glands exhibit circadian rhythmicity, but the interplay between the central circadian clock in the suprachiasmatic nucleus (SCN), the peripheral endocrine clock, and hormones is sparsely understood. We therefore studied the cellular localizations of the clock protein PER1, parathyroid hormone (PTH) and calcitonin (CT) in the parathyroid and thyroid glands, respectively. Thyroid glands, including the parathyroids, were dissected at different time-points from rats housed in 12 h:12 h light–darkness cycles, and were double-immunostained for PER1 and PTH or CT. Sera were analyzed for PTH, CT, phosphate, and calcium. In both glands, PER1 expression peaked late at night, while limited staining was seen during the daytime. High-resolution microscopy revealed cytosolic PER1 at zeitgeber time (ZT)12, and nucleic staining at ZT24 in both PTH and CT cells. PTH peaked at Z12–ZT16, while neither CT staining nor serum CT oscillated during the daily cycle. Serum PTH was significantly higher at ZT12 than ZT24, but only phosphate was found to exhibit significant diurnal oscillation. The staining of the calcium-sensitive receptor (CSR) did not demonstrate circadian oscillation. In conclusion, PER1 expression peaked late at night/early in the morning in hormone-producing cells of both the thyroid and parathyroid glands. In the parathyroids, this was preceded by a PTH peak, while neither CT nor CRS were found to oscillate.

## 1. Introduction

The circadian timing system consists of complex interactions between the central pacemaker located in the suprachiasmatic nucleus (SCN) of the hypothalamus and multiple peripheral clocks present in cells throughout the body. The daily need for alignment of the circadian clock with the day/night environment is primarily driven by light, but other cues such as food, activity, hormones, and sleep also participate in the fine-tuning of the circadian system [1,2,3,4].

Endogenous circadian clocks exist at all levels of the three major endocrine axes, namely the hypothalamic–pituitary–thyroid (HPT), the hypothalamic–pituitary–gonadal, and the hypothalamic–pituitary–adrenal axes. Neuronal signals from the SCN reach the neuroendocrine neurons in both the anterior hypothalamus and the hypothalamic paraventricular nucleus (PVN). The secretion of superior releasing hormones from these cells stimulates hormone secretion from the pituitary gland, which then affects the peripheral endocrine glands, causing them to produce the effector hormones [1,3,4]. The complex relationship between the endogenous clock and hormones at multiple levels is, however, only partly understood, making studies elucidating the cellular localization of clock components, the possible oscillation of hormones, and the phase of these, important.

The parathyroid glands, although in close proximity to the thyroid gland, are not controlled by the HPT axis; the parathyroid gland receives sympathetic, parasympathetic, and sensory innervation, which can be discriminated by the peptides they contain [5]. A functional circadian machinery has recently been revealed in the parathyroid glands of both rats and mice [6,7]. Although studied for decades, the results pertaining to the circulating PTH in both rodents and humans have been divergent, showing either no, biphasic, or significant diurnal rhythmicity with different peak times [8,9,10,11]. In addition to the possible circadian control of the PTH synthesis and/or secretion, the circulating PTH level is tightly regulated by ionized calcium. Thus, decreased amounts of blood calcium promote PTH secretion, which in turn lead to increased calcium concentrations [12,13]. Besides being a main regulator of calcium, PTH is an important but complex regulator of blood phosphate, which itself is also a regulator of PTH [14,15].

Calcium homeostasis is important, and the calcium concentration kept within narrow limits is detected by the calcium-sensing receptor (CSR), which, among others, is located on both parathyroid and calcitonin (CT) cells [12,13,16]. Binding calcium to CSR not only affects PTH but also CT secretion. The effect on the two secretions is opposite, meaning that high levels of calcium lead to enhanced blood CT [13,17,18], which in turn leads lowered calcium levels in rodents [13]. In addition, PTH regulates the level of circulating phosphate, and the secretion of PTH is, as mentioned, increased by phosphate [14,16].

Due to the many contradictory results on PTH, and lack of data on the cellular localization of clock components in both the parathyroid and thyroid glands, the aim of the study was to improve knowledge of the interrelationship between circadian clock components and the parathyroid gland. We revealed that in both the parathyroid and thyroid glands, the circadian clock is localized in the hormone-producing cells of the glands. The peak level of PER1 immunoreactivity in both glands was detected late at night (ZT20–ZT24, 12 h:12 h light–dark (LD), ZT0 = light on and ZT12 = light off), and the highest PTH expression was found at ZT12–ZT16, in both cells and sera. The PTH peak was preceded by the peak of significantly oscillating serum phosphate being maximal at midday to late in the day (ZT8), while neither calcitonin nor calcium exhibited diurnal rhythmicity.

## 2. Results

### 2.1. Diurnal Oscillation of PER1 in PTH Cells

Using immunohistochemistry, we evaluated the cellular localization and expression of PER1 in rat parathyroid glands at various timepoints during the daily cycle. As illustrated in Figure 1, the maximal PER1 expression was found late at night/dawn (ZT20–ZT2), while a very low level of PER1 staining was observed during the day (ZT6–ZT12), thus demonstrating a clear diurnal oscillation of PER1 protein in cells throughout the rat parathyroid gland.

We then aimed to evaluate the intracellular localization of PER1 by co-staining PER1 and the nuclear marker DAPI (4′,6-diamidino-2-phenylindole). We revealed dynamically changing PER1 expression during the 24 h LD cycle with a gradual increase in PER1 initially located in the cytoplasm early in the night (ZT12–ZT14, Figure 2b,c), followed by more intense nucleic staining at late night/early morning (ZT20–ZT2, Figure 2a,d).

Using triple immunostainings for PER1, PTH, and DAPI on tissue taken early in the night (ZT16), we next verified the presence of PER1 and PTH in identical cells. In Figure 3a, the cytoplasmatic PTH (in red) is shown to surround the PER1 nucleic staining (green). Figure 3b showing the DAPI staining in blue validates the nucleic localization of PER1 expression at late night/early morning, which now appears in turquoise.

The immunohistochemical staining of rat parathyroid glands shows that PER1 oscillates both in level and cellular localization across the daily cycle, being primarily cytoplasmatic late in the day/early in the night, while being nucleic late at night/early in the day. In addition, we have shown that the PTH-producing cells of the parathyroid glands contain a circadian clock.

### 2.2. Diurnal Cellular Oscillation of PTH

Due to the conflicting result published on PTH during the daily cycle, we evaluated both the PTH mRNA and protein through in situ hybridization (ISH) and immunohistochemistry, respectively, at various timepoints during the day. Figure 4a shows a graphic representation of semi-quantitative fluorescence ISH results of tissues dissected during 12 h:12 h LD cycles. Although not significant, the expression of PTH mRNA seems to peak early in the night (ZT12–ZT14). Figure 4b,c show representative microphotographs of PTH ISH of parathyroid glands taken at ZT10 (Figure 4b) and ZT14 (Figure 4c) showing low and high expression of PTH mRNA, respectively.

Representative results of the PTH immunostaining of parathyroid glands dissected during a 12 h:12 h LD cycle are shown in Figure 5, revealing diurnal oscillation of cellular PTH. Thus, clear hormone staining was found at the transition between day and night and early on in the night (ZT12–ZT16, Figure 5c,d), while very low/no staining is seen on pictures of tissues taken at the transition between night and day and early in the day (ZT24 and ZT4, Figure 5a,f).

### 2.3. Co-Localization of PER1 and Calcitonin

We and others have previously shown the oscillation of clock components in the thyroid gland [3,7,19]. However, whether circadian clock components are present in the calcitonin (CT)-producing C-cells of the gland have, to our knowledge, not been revealed. We therefore performed PER1 and CT double immunostaining. Figure 6 shows representative pictures with PER1 staining in green and CT in red. Especially at ZT24 (Figure 6f), when PER1 is localized in the cell, nuclei co-localization of the two in the same cells is evident, thus demonstrating the presence of PER1 in C-cells. In addition, Figure 6 illustrates that PER1 in C-cells displays the same oscillatory phase and cycle of intracellular localization as the PTH cells, thus being highly expressed in the nuclei at ZT24 (Figure 6f) while being absent in the cells at ZT8 (Figure 6b).

Since no obvious change in intensity of the calcitonin staining was revealed between the time points shown in Figure 6, we therefore evaluated the CT staining in more detail. Figure 7 solely illustrates the CT immunostaining, and although the intensity of the staining differs, no systematic changes in CT staining intensity during the 24 h LD cycle were observed (Figure 7).

### 2.4. The CSR Immunoreactivity in the Parathyroid and Thyroid Glands Shows No Diurnal Variation

The secretion of both PTH and CT is dependent on the CSR; we therefore investigated the localization and expression levels of CSR immunoreactivity during a full 12 h:12 h LD cycle. We found CSR immunoreactivity in the cell membranes throughout the parathyroid gland (Figure 8a,c–e) at all time points analyzed. CSR was also found in the cell membrane of cells in the thyroid gland at all time points (Figure 8d). The lack of any difference during the diurnal cycle is illustrated in Figure 8a,b taken at ZT10 and ZT20, respectively. Three-dimensional reconstruction of the localization of CSR in parathyroid cells is shown in Figure 8e. We did not observe diurnal oscillation of CSR in either the parathyroid or the thyroid gland (Figure 8a,b).

### 2.5. Serum Levels of PTH, CT, Phosphate and Calcium

After the cellular analyses performed using immunohistochemistry, we evaluated the serum concentrations of PTH (Figure 9a) and CT (Figure 9b) during full 12 h:12 h LD cycles. Neither of the two hormones oscillated significantly as a function of the 24 h cycle, *p* = 0.2 (PTH) and 0.4 (CT), respectively. However, the highest PTH concentration was seen at the transition between light and darkness (ZT12), whereafter it gradually declined, reaching nadir at ZT24; this resulted in a significantly higher concentration of PTH at ZT12 than at ZT24 (*p* < 0.05, Mann–Whitney).

The serum concentration of phosphate was found to display statistically significant 24 h sinusoidal oscillation, with a peak around ZT8 and nadir around ZT20. On the contrary, no difference in serum calcium was found during the 24 h LD cycle analyzed.

## 3. Discussion

The present study demonstrates the presence of the core clock protein PER1 in PTH-cells of parathyroid and in thyroid gland C-cells. Immunohistochemical examination of the parathyroid glands revealed a characteristic time course in both the level and intracellular localization of PER1 during a 12 h:12 h light/darkness cycle. Thus, from a just-detectable level during the day, PER1 protein accumulated within the cytoplasm at transition to night, after which PER1 was translocated to the nuclei when expressed at peak levels late at night/early morning. We have previously revealed an almost similar phase of oscillation and/or intracellular shuttling in the rat and mouse thyroid gland [7,19]. On the other hand, we found the phase of PER1 to be advanced in the ovary compared the thyroid and parathyroid glands [20], while in the adrenal gland it seems to be delayed [21], indicating some variation in the phases of different endocrine glands.

The parathyroid glands function as the major endocrine regulators of calcium–phosphate metabolism, and PTH plays a key role in both calcium and phosphate homeostasis [22]. To elucidate whether the clock in the parathyroid and thyroid gland could contribute to the timing of hormone synthesis and secretion involved in bone metabolism, we examined the cellular expression of PTH and calcitonin during a 12 h:12 h daily cycle. We found that cellular PTH displayed rhythmicity, being highest early in the night. In addition, a peak in circulating PTH at ZT12 was revealed. PTH has previously been shown to oscillate significantly in rats with maximal circulating levels at noon [10], and a recent study of young men kept under controlled conditions showed significant plasma PTH oscillation with a peak time at ZT22, corresponding to one hour before bedtime [11]; however, other studies of blood PTH have led to varying results [8,23]. The discrepancy between the results of these studies underscores the complexity of PTH regulation. In addition, differences in experimental conditions and the methods used for analysis might also have influenced the results. PTH secretion is closely regulated by the CSR located in the cell membrane of many cell types, including PTH and C-cells. CSR senses changes in the extracellular concentration of calcium ions [24], and even small changes in extracellular levels of ionized calcium rapidly de- or increase PTH secretion reciprocally to the change in ionized calcium [25], an effect mediated by the CSR [26]. Immunostaining of the thyroid and parathyroid glands for CSR did not reveal any diurnal oscillation.

Analysis of serum calcium did not show diurnal rhythmicity; however, the peak level of PTH at ZT12 coincides with the lowest concentration of calcium (Figure 9). Although we did not find any significant calcium variation, the highest concentration was detected during the day, which is in accordance with the findings of Milhaud et al. [27]. In agreement with the present study, no clear oscillation of blood calcium has previously been reported [12,28]. However, significant diurnal oscillation was recently reported [11]. As for calcium, we did not detect diurnal rhythmicity in CT (Figure 7 and Figure 9).

Serum phosphate is a function of many endogenous and exogenous factors, and the mechanisms for the interactions among these to the final circulating phosphate concentration are not fully clarified [14]. In spite of this, we found a statistically significant 24 h oscillation of the serum phosphate concentration at maximum during the day, a result in accordance with [27], and reciprocally to the oscillation found in humans [11]. One could speculate whether the diurnal changes in serum phosphate are related to the increase in PTH observed at ZT12.

Whether daily oscillations of PTH could be driven by the local clock in the parathyroid PTH cells has yet to be proven. However, oscillating PTH could act as a regulator of the endogenous clock on effector cells, as activation of the PTH receptor is able to enhance PER1 mRNA expression [29]. Many mechanisms are probably involved in the complex interaction between endocrine glands and the circadian clock.

As in every study, the present study has limitations as it is always possible to include more animals and analyses. More animals and samples may have resulted in significant 24 h oscillations of our PTH measurements. The model in this study was rats, and for some of the findings we have found similar results in mice. However, the generalizability to other species has not been shown.

## 4. Materials and Methods

### 4.1. Animals

Adult Wistar rats of both sexes (Taconic Breeding Centre, Lille Skensved, Denmark) weighing 150–200 g were housed under standard laboratory conditions with ad libitum access to food and water. In total, 128 animals were used, which were entrained before experiments for at least 14 days to 12 h:12 h LD cycles. All animals were treated according to the principles of Laboratory Animal Care (Law on Animal experiments in Denmark, publication 1306, 23 November 2007).

### 4.2. Immunohistochemistry

For immunohistochemistry, four animals were transcardially perfusion-fixed with Stefanini’s fixative at each of the following time points: ZT2, ZT4, ZT6, ZT10, ZT12, ZT14, ZT16, ad ZT18, and ZT24, whereafter the thyroid glands including the parathyroid glands were dissected and stored at −80 °C until further processing. The thyroid glands, including the parathyroid glands, were cut on a cryostat as 12 µm thick sections, and processed for PER1 immunohistochemistry as previously described, using PER1 antiserum raised and characterized in our own laboratory [20] and counterstained using 4′, 6′-diamidino-2-phenylindole (DAPI). PTH staining was carried out using goat polyclonal antibodies (N-18: sc-9676, Santa Cruz Biotechnology, Dallas, TX, USA), rabbit polyclonal antibodies (T-4235, Peninsula Laboratories, San Carlos, CA, USA) were used for CT, and mouse monoclonal antibodies (19347, Abcam, Cambridge, UK) were used for CSR. The CSR staining sections were pretreated using antigen retrieval solution (DAKO Ph 6 for 2 × 7 min at 100 °C). Three to four sections from each animal were stained in each staining using the procedure described previously [30]. Fluorescence images were obtained using an iMIC confocal microscope (TILL Photonics, FEI, Münich, Germany) and appropriate filter settings for the detection of Alexa 488 and DAPI. Images were edited for contrast and brightness in Fiji or in Adobe Photoshop (Adobe Systems, San Jose, CA, USA) and combined into plates using Adobe Illustrator (Adobe Systems).

### 4.3. In Situ Hybridisation Histochemistry

For fluorescence ISH, three animals were decapitated at each of the following time points: ZT2, ZT4, ZT6, ZT10, ZT12, ZT14, ZT16, ZT18, and ZT24; the thyroid complex was removed and immediately frozen on dry ice. The dissected tissue was cut into 12 µm thick sections on a cryostat with 3–5 sections on each glass.

The detection of PTH mRNA was carried out by ISH using digoxigenin-labeled antisense RNA probes as previously described [31]. The template used for the in vitro transcription was a plasmid (304) containing the coding region of mouse PTH cDNA (nt. 83–479 of BC099456) in pBluescript. Before transcription with T7- or T3-polymerase for antisense and sense RNA, respectively, the plasmid was linearized with either HindIII (antisense template) or XbaI (sense template).

### 4.4. ELISA

For hormone analysis, trunk blood was obtained every 4 h after decapitation at ZT4, ZT8, ZT12, ZT16, ZT20, and ZT24. The blood samples were allowed to clot at room temperature for 30 min prior to centrifugation at 1000× *g*. The removed supernatants were stored at –20 °C until they were assayed. Serum concentrations of intact PTH (1–84) were measured by a two-site enzyme-linked immunosorbent assay for use in rats (60–2500, Immutopics, San Clemente, CA, USA). The interassay coefficient of variation of the assay was 5.1%, and the minimum detectable concentration was 1.6 pg/mL. Serum concentrations of CT were measured by a sandwich enzyme immunoassay for use in rats (CSB-E05132r, Cusabio, Houston, TX, USA). The interassay coefficient of variation was 10%, and the minimum detectable concentration was 0.4 pg/mL. The concentrations of calcium and phosphate were determined on Cobas 8000 (Roche Diagnostics, Copenhagen, Denmark) with assays having interassay coefficients of variation of 2.5% and 6.0%, and minimum detectable concentrations of 0.2 mM and 0.1 mM, respectively.

### 4.5. Statistical Analysis

Values are presented as the means ± standard error of the mean (SEM). Diurnal changes in PTH, CT, calcium, and phosphate concentrations were analyzed using the methods for cosinor-based rhythmometry as described by Nelson et al. [32]. The data were thus fitted to a combined cosine and sine function as follows: PTH = M + k1Cos (2πt⁄24) + k2SIN (2πt⁄24). Substituting Cos (2πt⁄24) = C and SIN (2πt⁄24) = Z gives the following expression: PTH = M + k1 C + k2Z. The model fit was then tested using the general linear model procedure in the SAS statistical software package version 9.4 (Cary, NC, USA). *p* < 0.05 was considered statistically significant.

## Figures and Tables

**Figure 1 ijms-25-13006-f001:**
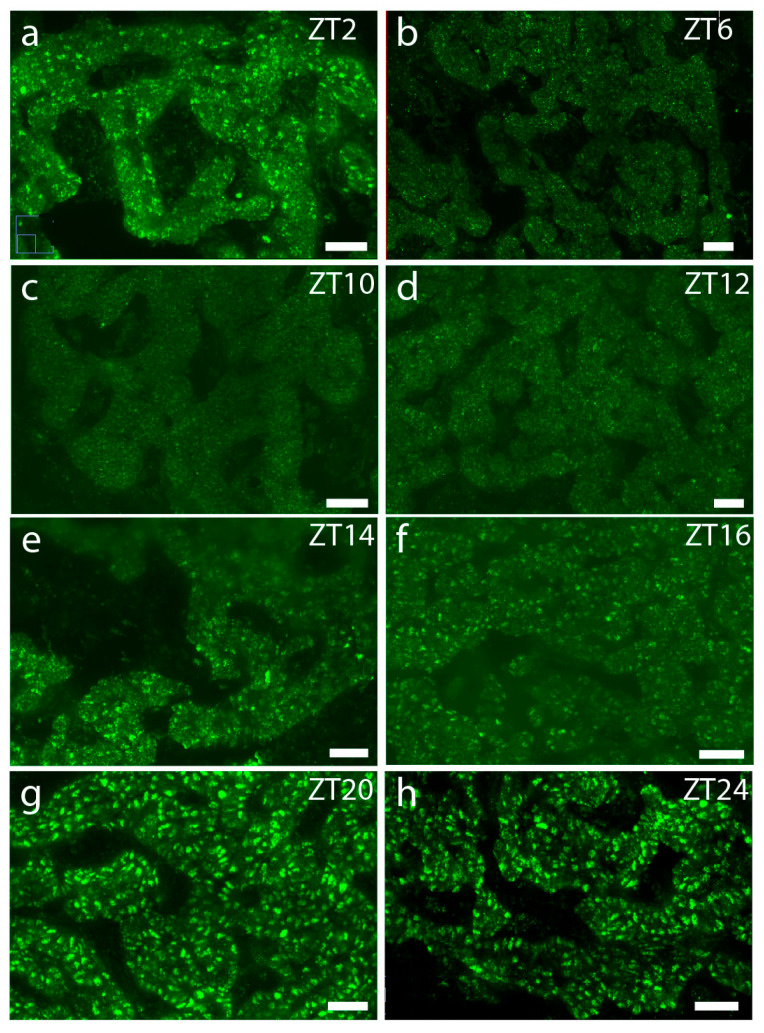
Daily oscillation of PER1 immunostaining in rat parathyroid glands. Confocal images of PER1 staining of glands taken at different time points during a 12 h:12 h light–dark cycle. Glands of four animals at each timepoint were stained, and representative pictures are shown. (**a**) The presence of PER1 in a parathyroid gland taken at ZT2; (**b**) at midday (ZT6), very little PER1 staining is seen; (**c**,**d**) there is almost an absence of staining in glands from ZT10 and ZT12; (**e**,**f**) at ZT14 and ZT16, more evident staining is appearing; (**g**,**h**) intense PER1 staining in parathyroids from ZT20 and ZT24. Scale bars: 25 µm.

**Figure 2 ijms-25-13006-f002:**
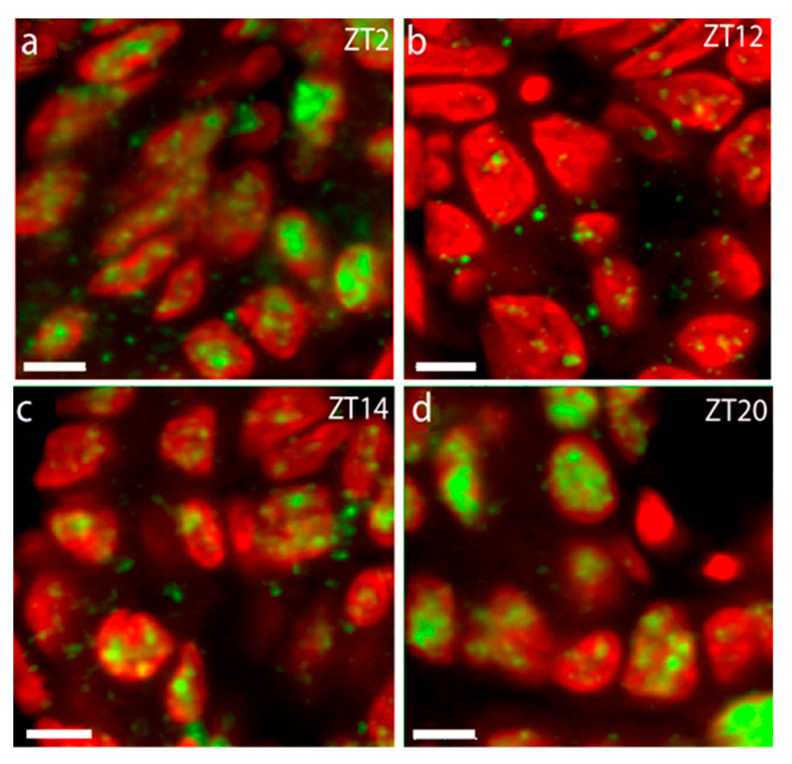
Intracellular distribution of PER1 rat parathyroid glands during the daily cycle. High-power confocal images of PER1 immunostaining (green) and DAPI nuclear staining (red) at selected time points during a 12 h:12 h light–dark cycle. (**a**) At ZT2, PER1 is primarily localized in the cell nuclei; (**b**) the primarily cytoplasmic PER1 localization at ZT12; (**c**) at ZT14, PER1 is seen to appear in the nuclei; (**d**) at ZT20, PER1 is almost entirely present in the cell nuclei. Scale bars: 10 µm.

**Figure 3 ijms-25-13006-f003:**
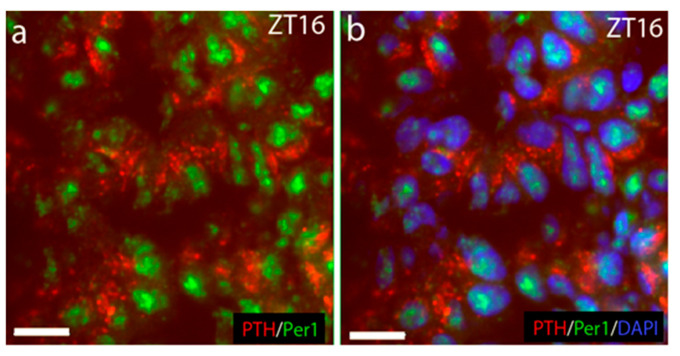
Co-localization of PER1 and PTH in the same cells. (**a**) Immunostaining of PER1 in green and PTH in red in the same cells; (**b**) DAPI staining in blue, confirming the nucleic localization of PER1, now appearing in turquoise. Scale bars: 10 µm.

**Figure 4 ijms-25-13006-f004:**
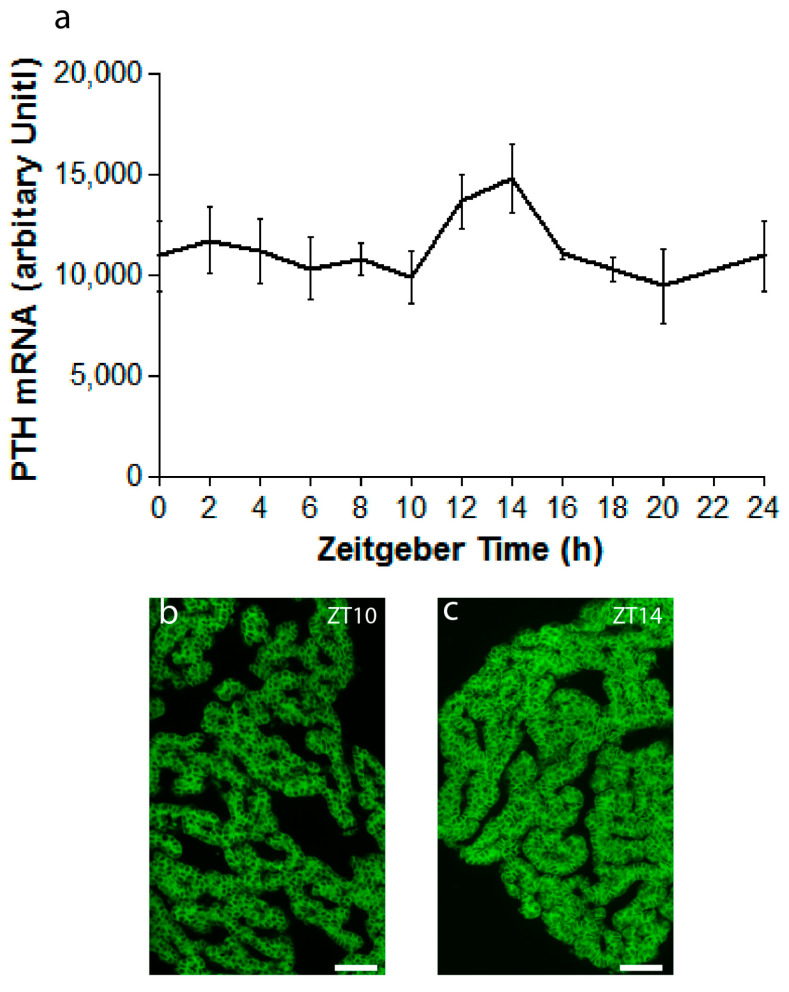
Results of in situ hybridization (ISH) detecting PTH mRNA during the 12 h:12 h light–dark cycle. (**a**) Graphical illustration of the semi-quantification of ISH results showing higher PTH mRNA expression at the beginning and early in the night (ZT12-ZT14) than late at night (n = 2–4); the difference was, however, not statistically significant; (**b**) representative photomicrograph of PTH ISH at ZT10; (**c**) ISH of PTH mRNA at ZT14. Scale bars: 25 µm.

**Figure 5 ijms-25-13006-f005:**
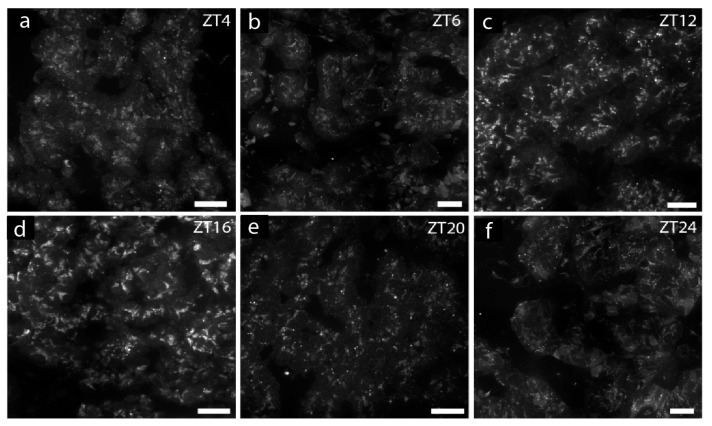
Confocal images of immunofluorescent PTH staining of rat parathyroid glands taken during a 12 h:12 h light–dark cycle. (**a**,**b**): No or very low staining in early morning and midday; (**c**,**d**): Intense staining in early night; (**e**,**f**): Decreasing staining in late night. Scale bares: 25 µm.

**Figure 6 ijms-25-13006-f006:**
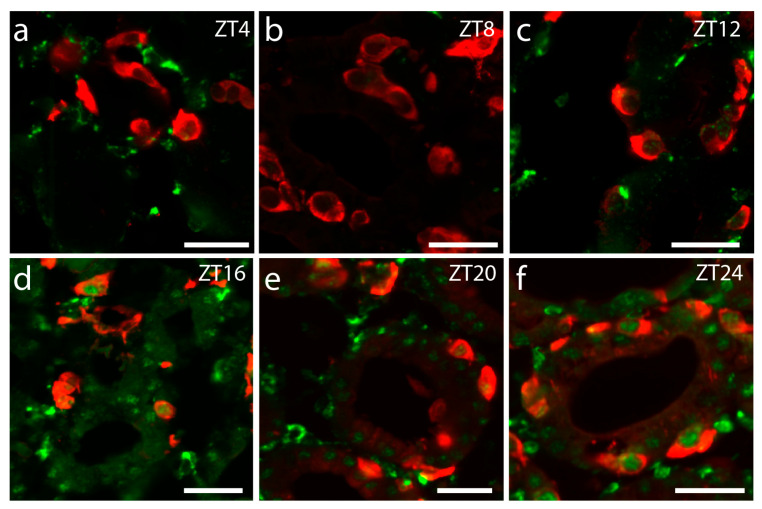
Confocal images of PER1 (green) and calcitonin (red) double immunofluorescence staining of the thyroid gland taken during a 12 h:12 h light–dark cycle. PER1 is seen to vary in both intensity and localization during the daily cycle. (**b**) shows absence in the day; (**d**–**f**) increasing PER1 staining in the night; (**f**) Shows nucleic localization at dawn. Scale bars: 25 µm.

**Figure 7 ijms-25-13006-f007:**
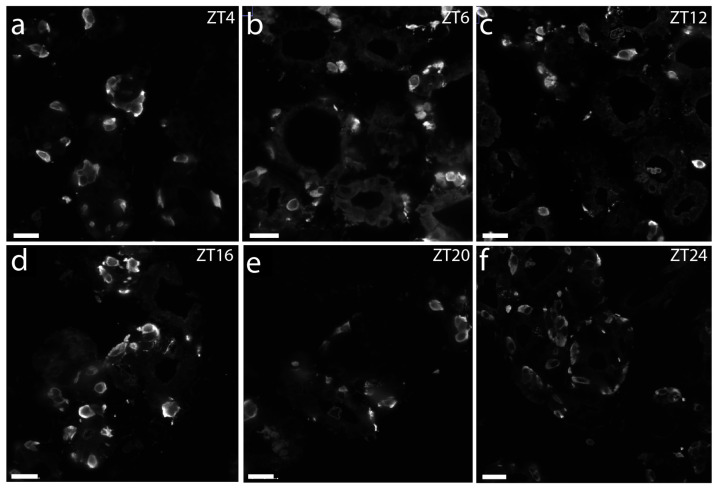
Confocal images of calcitonin immunostaining of thyroid glands taken during a 12 h:12 h light–dark cycle. Although differing, diurnal oscillation in calcitonin staining intensity was not found. Scale bares: 25 µm.

**Figure 8 ijms-25-13006-f008:**
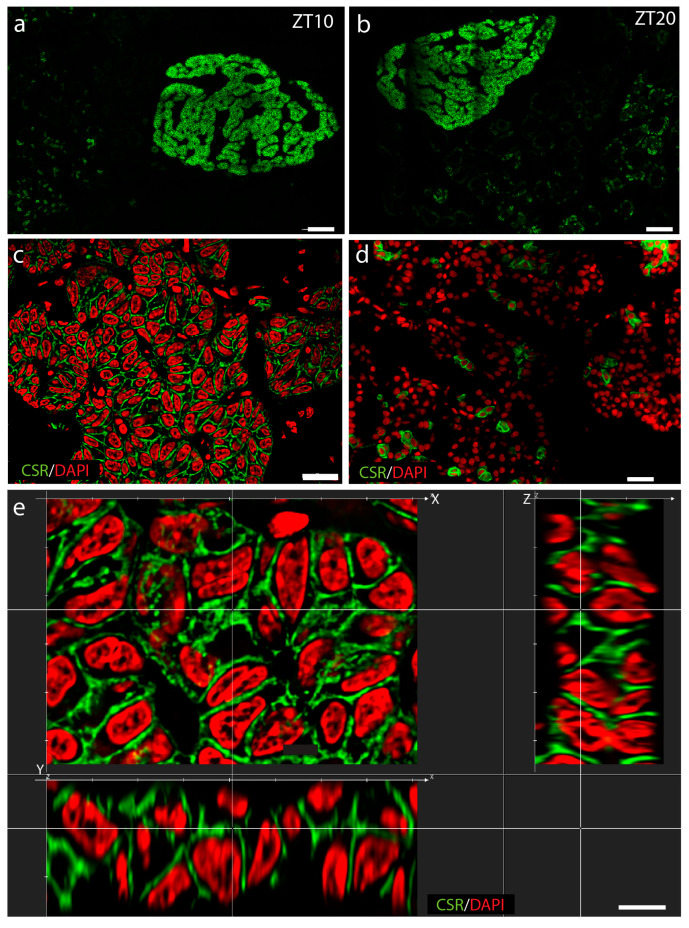
Confocal microphotographs of CSR immunostaining of the parathyroid and thyroid glands. (**a**) Picture of both glands taken at ZT10; (**b**) picture of both glands taken ZT20; (**c**) higher magnification of the parathyroid gland showing CSR in green and DAPI in red; (**d**) higher magnification of CSR (green) and DAPI (red); (**e**) detailed high-resolution analysis in and 3D reconstruction (x, Y, Z) showing the cell membrane localization of CSR. Scale bars: (**a**,**b**): 50 µm; (**c**): 20 µm; (**d**): 25 µm; (**e**): 8 µm.

**Figure 9 ijms-25-13006-f009:**
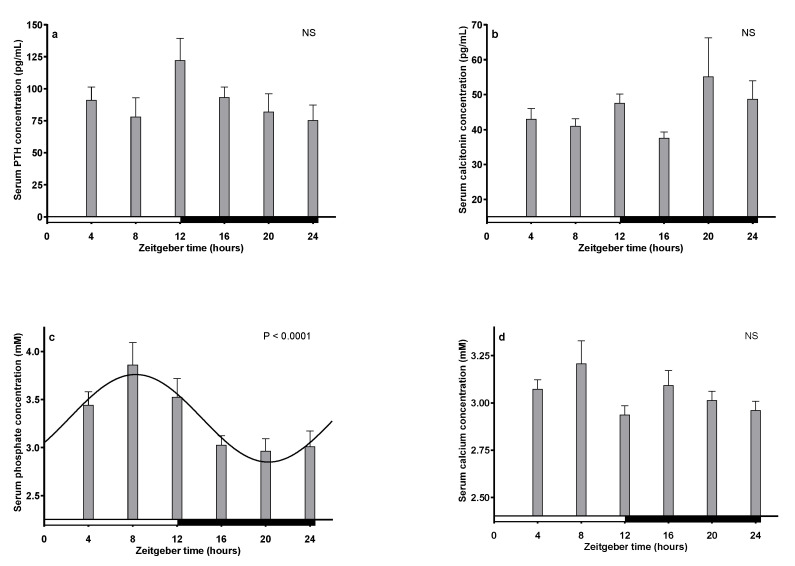
Graphical illustration of serum concentrations of PTH (**a**), calcitonin (**b**), phosphate (**c**), and calcium (**d**) during a 12 h:12 h LD cycle. PTH and calcitonin were measured by commercial ELISA kits, and phosphate and calcium concentrations were measured on Cobas 8000. Only the concentration of phosphate displayed significant 24 h sinusoidal oscillation. The concentration of PTH at ZT12 was, however, significantly higher (*p* < 0.05) than at ZT24. NS: Not Significant.

## Data Availability

The raw data supporting the conclusions of this article will be made available by the authors on request.

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
