# Peer review of "PER1 Oscillation in Rat Parathyroid Hormone and Calcitonin Producing Cells"

_ijms, 2024, doi:10.3390/ijms252313006_

Round 1

Reviewer 1 Report

Comments and Suggestions for Authors

The manuscript of Dr. Georg, Dr. Jorgensen, and Dr. Hannibal is proposing a demonstration of the oscillation of protein PERIOD-1 by immunofluorescent labeling of cells from thyroid and parathyroid glands. Immunolabeling of cryostat sections were performed in quadruplicate on male and female adult Wistar rats over 24 hours, every 4 hours. The strong points of the study are that authors are following the expression of PERIOD1, Calcium sensitive receptor, parathyroid hormone, calcitonin, in parathyroid and thyroid glands. They claim first demonstration of the presence of PERIOD1 in C-cells of the glands using double labeling with Calcitonin.

The main weak point is that the demonstration is only qualitative on the intensity of labelling of representative pictures. Why not presenting the semi-quantitation on averaged quadruplicate (see (3) for proposal) ?

The manuscript needs improvements before publication.

Main modifications

(1) In the Material and Method section, you have indicated using 4 rats per time point. The reader infers that you have used 24 male rats and 24 female rats to realize the study. Thanks to state clearly this fact in the section.

(2) How many sections did you stained per animal?

(3) The photos can be clearly used to propose some quantitative assessment of PERIOD1 using a free image processing software like ImageJ. The manuscript will be greatly improved by providing semi-quantification of PERIOD1 like the semi-quantitative measurements shown on Figure 4a for PTH mRNA.

(4) The phosphate oscillations presented on Figure 9c are in favor of daily rhythm. The situation for PTH on panel A is described as non-sinusoidal but with significant higher expression at ZT-12 comparatively to ZT-24. Could you justify your point of view? From my point of view, the sinusoidal oscillation is possible with PTH. Did you realize statistical analysis of circadian rhythms?

(5) The Statistical analysis section is reduced to the analyses of PTH, CT, Calcium, phosphate by the method of Nelson. It is absolutely not obvious to reduce treatment of PERIOD1 labeling to a qualitative analysis. Even if the outcome is not significant, the discussion about the semi-quantitative approach of PERIOD1 has to be improved.

Minor Modifications

(1) In the abstract, please clarify the irrelevant word «only phosphate was the found to exhibit significant diurnal oscillations». Maybe« the» can be removed?

(2) Please use bigger characters for Figure 9, many readers will have difficulty to localize a, b, c, d panels on present illustration.

Author Response

Comments and Suggestions for Authors, Reviewer 1:

The manuscript of Dr. Georg, Dr. Jorgensen, and Dr. Hannibal is proposing a demonstration of the oscillation of protein PERIOD-1 by immunofluorescent labeling of cells from thyroid and parathyroid glands. Immunolabeling of cryostat sections were performed in quadruplicate on male and female adult Wistar rats over 24 hours, every 4 hours. The strong points of the study are that authors are following the expression of PERIOD1, Calcium sensitive receptor, parathyroid hormone, calcitonin, in parathyroid and thyroid glands. They claim first demonstration of the presence of PERIOD1 in C-cells of the glands using double labeling with Calcitonin.

We appreciate the reviewer for the comments and the constructive points raised to improve the manuscript. Below we have answered the questions and addressed the points raised and accordingly modified the manuscript, which, hopefully, is now ready for publication.

The main weak point is that the demonstration is only qualitative on the intensity of labelling of representative pictures. Why not presenting the semi-quantitation on averaged quadruplicate (see (3) for proposal) ?

The manuscript needs improvements before publication.

Thank you very much for raising the point, which we ourselves have considered during the study. The goal of the study, however, was to reveal whether the hormone producing cells of the parathyroid and thyroid gland were identical to the clock cells of the glands. This was done by co-localization using IHC, which primarily is a qualitative method. In addition, we aimed to reveal the intracellular dynamic of PER1. The strength of IHC is its ability to localize proteins within specific cells, and it is not especially suitable for quantitative purposes which always must be done with great care. On the contrary, we have previously been able to quantitate ISH with good accuracy. As evaluation of PTH included both ISH and IHC, we quantified the ISH as shown in Figure 4a. Regarding PER1, the present and our previous work have shown identical acrophase of clock proteins in the thyroid and parathyroid glands, and previously we have used qPCR to quantify the acrophase of various clock mRNAs (Per1, Per2 and Bmal1) in both rat and mouse in thyroid glands. The PER1 staining in the present study was, as previously mentioned, primarily used to evaluate whether the clock and hormone producing cells of the thyroid and parathyroid glands, respectively, are identical and in addition, to reveal the intracellular dynamics of PER1 in the parathyroid gland.

Main modifications

  • In the Material and Method section, you have indicated using 4 rats per time point. The reader infers that you have used 24 male rats and 24 female rats to realize the study. Thanks to state clearly this fact in the section.

For immunocytochemistry, 4 rats of mixed sex at 9 timepoints were used giving in total 36 animals. In addition, blood samples of 12 -14 animals of mixed sex at 6 timepoints were used for ELISAs giving 74 animals. For ISH, 3 animals at 9 timepoints were examined, as some of these were identical to the ones used for ELISAs, an additional 18 rats were used. The total number of animals (128) used is now specified in Materials and Methods (p. 10, l. 270).

  • How many sections did you stained per animal?

Three to five sections were stained per animal. This information has now been included in the manuscript (p. 11, l. 286 and l. 297).

  • The photos can be clearly used to propose some quantitative assessment of PERIOD1 using a free image processing software like ImageJ. The manuscript will be greatly improved by providing semi-quantification of PERIOD1 like the semi-quantitative measurements shown on Figure 4a for PTH mRNA.

We acknowledge that in some instances it may be relevant to quantify immunohistochemistry. However, as stated above, it was not the purpose of the present study to evaluate in details oscillation of clock components as this have been done before, but rather to evaluate whether the hormones producing cells were identical to the clock cell of the glands, in addition to reveal the intracellular dynamics of PER1 in the parathyroid gland.

(4) The phosphate oscillations presented on Figure 9c are in favor of daily rhythm. The situation for PTH on panel A is described as non-sinusoidal but with significant higher expression at ZT-12 comparatively to ZT-24. Could you justify your point of view? From my point of view, the sinusoidal oscillation is possible with PTH. Did you realize statistical analysis of circadian rhythms?

We analyzed all the data shown in figure 9 with sinusoidal statistics as first choice. For PTH it gave P = 0.15. We therefore chose to compare the data obtained at ZT12 and ZT24 using the Mann-Whitney test, which gave P = 0.036.

(5) The Statistical analysis section is reduced to the analyses of PTH, CT, Calcium, phosphate by the method of Nelson. It is absolutely not obvious to reduce treatment of PERIOD1 labeling to a qualitative analysis. Even if the outcome is not significant, the discussion about the semi-quantitative approach of PERIOD1 has to be improved.

Please see our answers above.

Minor Modifications

  • In the abstract, please clarify the irrelevant word «only phosphate was the found to exhibit significant diurnal oscillations». Maybe« the» can be removed?

The reviewer is perfectly correct that the “the” is a mistake, which now has been removed.

(2) Please use bigger characters for Figure 9, many readers will have difficulty to localize a, b, c, d panels on present illustration.

We fully agree, and the letters a, b, c, and d have now been enlarged.

Reviewer 2 Report

Comments and Suggestions for Authors

 The scientific topic is interesting. The authors studied cellular localization of the clock protein PER1, parathyroid hormone (PTH) and calcitonin (CT) in parathyroid and thyroid glands, respectively.

Are the parathyroid glands controlled by a superior “hypothalamic-pituitary-axis?" Please include this fact that it is not controlled by HPT axis.

Does the circadian clock control the genes expressed in peripheral tissues?

What is the cell turnover rate in the parathyroid gland under normal conditions?

Does disruption of the circadian rhythm cause abnormal cell growth?

Please add the facts that parathyroids were dissected every 4th h in the methods section of the paper. If possible, please cite a reference.

Were the animals exposed to any particular light during the experiments? Were the eyes covered during the procedure of anesthesia?

PTH secretion exhibits a diurnal variation with acrophase during the inactive period in both humans and rodents. The authors should address this point.

The circadian clockwork is entrained by which environmental cues? Can any feeding signal or light input influence the results?

According to the present study, when calcium should be given in patients with renal hyperparathyroidism? Is it morning or evening?

Can acute hypercalcemia suppress the pulsatile secretion component?

The autonomic innervation of the parathyroids may be added.

Limitations of the study should be mentioned.

Author Response

The scientific topic is interesting. The authors studied cellular localization of the clock protein PER1, parathyroid hormone (PTH) and calcitonin (CT) in parathyroid and thyroid glands, respectively.

Are the parathyroid glands controlled by a superior “hypothalamic-pituitary-axis?" Please include this fact that it is not controlled by HPT axis.

We thank the reviewer for his comment, The fact that the parathyroid gland is not controlled by the HPT is stated in third paragraph of the introduction (p. 45 – 46).

Does the circadian clock control the genes expressed in peripheral tissues?

The central circadian clock of the SCN both directly and indirectly affects the gene expression in peripheral tissue. Both the central circadian clock and the peripheral clocks present in all organs affect the gene expression in these peripheral tissues (Mohawk et al. 2012, Ann. Rev. Neurosci. 35:445-62).

What is the cell turnover rate in the parathyroid gland under normal conditions?

Cellular turnover rate is very important and interesting in many physiological aspects. However, it is not an issue dealt with in the present study.

Does disruption of the circadian rhythm cause abnormal cell growth?

The short answer is yes there is a connection between cell growth and the circadian system. However, this is as regulation of many other physiological and cellular events, dependent on the conditions. The connection between for example cancer and circadian rhythms is a hugely expanding field with an increasing number of studies being published. The aim of the present study was however not to evaluate neither cell turnover rate nor abnormal cell growth.

Please add the facts that parathyroids were dissected every 4th h in the methods section of the paper. If possible, please cite a reference.

We apologize if it is unclearly stated in the manuscript, however under both IHC and ISH, the zeitgeber times (ZTs) of the animals used are stated (p. 10, l. 276 - 277 and p. 11, l. 295), the times for tissue dissection was not exactly every 4th h, as we dissected the glands at 9 different time-points. For serum sampling, however, these were taken every 4th h, which is now stated in the “4.4 ELISA” (p. 11, l. 305 - 306) in addition to the ZTs.

Were the animals exposed to any particular light during the experiments? Were the eyes covered during the procedure of anesthesia?

As stated at p. 10, l. 270 animals were exposed to 12:12 light-dark cycles for at least 14 days before experiments. For animals taken at ZT2 – ZT12 (ZT0 = light on, ZT12 = light off), the anesthesia/decapitation was done in light. Animals taken at ZT14 – ZT24 were anesthetized/decapitated, and organs dissected/serum sampled in red light, which does not influence the circadian rhythm. The animals taken at night-time (ZT12 – ZT24) were thus not exposed to white light neither during anesthesia/decapitation nor sample sampling.

PTH secretion exhibits a diurnal variation with acrophase during the inactive period in both humans and rodents. The authors should address this point.

Multiple reports of the diurnal variation in circulating PTH in both humans and rodents exists, and we have discussed the subject in the second paragraph of the Discussion (p.9, l. 229 – 237).

The circadian clockwork is entrained by which environmental cues? Can any feeding signal or light input influence the results?

As stated in the first paragraph of the Introduction, light is the prime zeitgeber of the circadian clock, however, other cues exist such as food, activity, hormones, and sleep (p. 1, l. 31)

According to the present study, when calcium should be given in patients with renal hyperparathyroidism? Is it morning or evening?

We are not aware that calcium should belong to “chronotherapeutic” medication, but it is an interesting thought.

Can acute hypercalcemia suppress the pulsatile secretion component?

We do not think your question has been tested in the clinic.

The autonomic innervation of the parathyroids may be added.

We have described the innervation of the thyroid and parathyroid gland in details in our previous work (Fahrenkrug and Hannibal 2011, Gen Comp Endocrinol 171:105-13) and have now added the following sentence: “the parathyroid gland receives both sympathetic, parasympathetic and sensory innervation, which can be discriminated by the peptides they contain (5).” in the present manuscript (p. 1, l. 44 – 46).

Limitations of the study should be mentioned.

Thank you for this comment as we obviously lack to mention the limitations of the study. This have now been included with the following: “As in every study, the present has limitations as it is always possible to include more animals and analyses. More animals and samples may have resulted in significant 24-h oscillations of our PTH measurements. The model in this study was rats and for some of the findings we have found similar results in mice. However, the generalizability to other species has not been shown.” (p. 10, l.261-265).

Round 2

Reviewer 1 Report

Comments and Suggestions for Authors

The manuscript of Dr. Georg, Dr. Jorgensen, and Dr. Hannibal is proposing a demonstration of the oscillation of protein PERIOD-1 by immunofluorescent labeling of cells from thyroid and parathyroid glands. This first demonstration of the presence of PERIOD1 in C-cells of the glands using double labeling with Calcitonin deserves publication. The authors have correctly addressed all comments.